EMBO
Molecular Medicine

# CRTH2 promotes endoplasmic reticulum stress-induced cardiomyocyte apoptosis through m-calpain

Shengkai Zuo[1,2], Deping Kong[1], Chenyao Wang[2] , Jiao Liu[2], Yuanyang Wang[1], Qiangyou Wan[2], Shuai Yan[2], Jian Zhang[1], Juan Tang[2], Qianqian Zhang[2], Luheng Lyu[2,3], Xin Li[1], Zhixin Shan[4], Li Qian[5], Yujun Shen[1,*] & Ying Yu[1,2,**]

## Abstract

Apoptotic death of cardiac myocytes is associated with ischemic heart disease and chemotherapy-induced cardiomyopathy. Chemoattractant receptor-homologous molecule expressed on T helper type 2 cells (CRTH2) is highly expressed in the heart. However, its specific role in ischemic cardiomyopathy is not fully understood. Here, we demonstrated that CRTH2 disruption markedly improved cardiac recovery in mice postmyocardial infarction and doxorubicin challenge by suppressing cardiomyocyte apoptosis. Mechanistically, CRTH2 activation specifically facilitated endoplasmic reticulum (ER) stress-induced cardiomyocyte apoptosis via caspase-12-dependent pathway. Blockage of m-calpain prevented CRTH2-mediated cardiomyocyte apoptosis under ER stress by suppressing caspase-12 activity. CRTH2 was coupled with $G_{\alpha q}$ to elicit intracellular $Ca^{2+}$ flux and activated m-calpain/caspase-12 cascade in cardiomyocytes. Knockdown of caspase-4, an alternative to caspase-12 in humans, markedly alleviated CRHT2 activation-induced apoptosis in human cardiomyocyte response to anoxia. Our findings revealed an unexpected role of CRTH2 in promoting ER stress-induced cardiomyocyte apoptosis, suggesting that CRTH2 inhibition has therapeutic potential for ischemic cardiomyopathy.

**Keywords** calpain; cardiomyocyte apoptosis; CRTH2; endoplasmic reticulum stress; prostaglandin $D_2$

**Subject Categories** Cardiovascular System; Vascular Biology & Angiogenesis

## Introduction

Cardiovascular diseases continue to be the leading cause of morbidity and mortality worldwide (Benjamin *et al*, 2017). Heart failure, characterized by the inability of the ventricle to sufficiently pump blood, is the end stage of various forms of cardiovascular diseases, such as myocardial infarction (MI), valvular heart disease, and cardiomyopathy (Harjola *et al*, 2016). Progressive cardiomyocyte loss is a key pathogenic factor in the development of heart failure. Based on morphological manifestations, three distinct types of cell death, namely apoptosis, necrosis, and possibly autophagy, occur in cardiac myocytes during MI and heart failure (Lee & Gustafsson, 2009; Whelan *et al*, 2010). While signal transduction pathways involved in cell death have been widely investigated, how cardiac myocytes initiate specific death signaling in response to different stresses, such as ischemia and toxic chemicals, remains unclear.

Apoptosis is a process of programmed cell death that plays an important role in the progression of heart failure (Lee & Gustafsson, 2009). Two classic pathways—the extrinsic and intrinsic pathways—mediate apoptotic signaling in mammalian cells (Moe & Marin-Garcia, 2016). The extrinsic apoptosis is triggered by death ligands, such as Fas and tumor necrosis factor-α. The binding of the ligands and their individual death receptors leads to activation of caspase-8-mediated apoptotic cascade. The intrinsic pathway, also called the mitochondrial apoptosis pathway, is initiated by intracellular stress, such as oxidative stress or DNA damage, which ultimately results in mitochondrial membrane permeabilization, cytochrome C release, and subsequent activation of caspase-9-mediated apoptotic cascade. Caspase-12 is an endoplasmic reticulum (ER) resident caspase that has been recently identified to mediate ER stress-induced apoptosis, such as high calcium concentration or low oxygen (Nakagawa & Yuan, 2000; Nakagawa *et al*, 2000). Ischemia and doxorubicin

1   Department of Pharmacology, Key Laboratory of Immune Microenvironment and Disease (Ministry of Education), School of Basic Medical Sciences, Tianjin Medical University, Tianjin, China
2   Key Laboratory of Food Safety Research, Institute for Nutritional Sciences, Shanghai Institutes for Biological Sciences, Chinese Academy of Sciences, Shanghai, China
3   Department of Biology, University of Miami College of Arts and Science, Miami, FL, USA
4   Medical Research Department of Guangdong General Hospital, Guangdong Cardiovascular Institute, Guangdong Academy of Medical Sciences, Guangzhou, Guangdong, China
5   McAllister Heart Institute, University of North Carolina at Chapel Hill, Chapel Hill, NC, USA
    *Corresponding author. Tel/Fax: +86 22 83336668; E-mail: yujun_shen@yahoo.com
    **Corresponding author. Tel/Fax: +86 22 83336668; E-mail: yuying@tmu.edu.cn

(DOX) can increase ER stress and apoptosis in the heart (Lam et al, 2013; Xu et al, 2015; Fu et al, 2016). Apoptotic cardiomyocytes are observed in cardiac tissues from patients with MI, dilated cardiomyopathy, and heart failure (Narula et al, 1996; Olivetti et al, 1996; Saraste et al, 1997), as well as in animal models of different cardiac injuries (Fliss & Gattinger, 1996; Qin et al, 2005). Pharmacological and genetic inhibition of cardiomyocyte apoptosis diminishes infarct size and improves cardiac function after MI (Whelan et al, 2010). Therefore, targeting apoptosis is a promising preventive and therapeutic strategy for heart failure (Yang et al, 2013a).

Prostaglandin (PG) $D_2$ is a bioactive metabolite of arachidonic acid produced through the sequential reaction of cyclooxygenases (COXs) and $PGD_2$ synthases. $PGD_2$ exerts its functions through activation of the D-prostanoid receptor 1 (DP1) and the chemoattractant receptor-homologous molecule expressed on T helper type 2 cells (CRTH2, also named as DP2). It has been implicated in various pathophysiological events, especially inflammation (Santus & Radovanovic, 2016). DP1 receptor is abundantly expressed in brain tissues, mast cells, and macrophages, and CRTH2 is highly expressed in immune cell-enriched organs and the heart (Sawyer et al, 2002; Santus & Radovanovic, 2016). Since DP1 receptor is not detectable in cardiac tissues (Katsumata et al, 2014), $PGD_2$/DP1 axis mediates glucocorticoid-induced cardioprotection against ischemia (Tokudome et al, 2009), probably through M2 macrophage-mediated timely resolution of inflammation in injured hearts (Kong et al, 2016, 2017). However, the role of CRTH2 in cardiac recovery from ischemia is unknown.

In the present study, we observed that $PGD_2$/CRTH2 axis was markedly upregulated in cardiomyocytes in response to anoxia and DOX treatment, which increased ER stress in cardiomyocytes. Unexpectedly, CRTH2 deletion attenuated anoxia or DOX-induced apoptosis in cardiomyocytes and conferred cardioprotection against MI and DOX treatment in mice. CRTH2 deficiency suppressed ER-specific caspase-12 activation in cardiomyocytes by reducing $Ca^{2+}$-dependent cysteine protease m-calpain activity in mice. In human cardiomyocytes, CRTH2 activation promoted anoxia-induced apoptosis through activating human caspase-4, an ER caspase homolog to mouse caspase-12. Thus, our results demonstrated that CRTH2 facilitated ER stress-induced apoptosis through the m-calpain/caspase-12 signaling pathway.

# Results

## PGD$_2$/CRTH2 axis is upregulated in cardiomyocytes in response to anoxia

To determine whether PGs are involved in anoxia-induced apoptosis in cardiomyocytes, we isolated primary cardiomyocytes from neonatal mice and examined the alterations of PG production and PG receptor expression in cardiomyocytes in response to anoxia. $PGD_2$, $PGE_2$, $PGF_{2\alpha}$, and $TxB_2$ were significantly elevated in response to anoxia (Fig 1A and Appendix Fig S1A). CRTH2, IP, FP, EP1, and EP4 receptors were abundantly expressed in cardiomyocytes, whereas DP1 was barely detected. Anoxia boosted CRTH2 expression (2.7-fold), while repressed FP and EP1 expression (Fig 1B and Appendix Fig S1B). To investigate whether the changes in CRTH2, FP, or EP1 receptor contribute to anoxia-induced apoptosis in

cardiomyocytes, we directly examined the activation of caspase-3, the main effector caspase, in these receptor agonist-treated cardiomyocytes. The CRTH2 agonist DK-$PGD_2$ markedly induced caspase-3 activation in cardiomyocytes, but the other agonists (Lat-FA, FP agonist; ONO-DI-004, EP1 agonist; misoprostol, EP4 agonist) had no overt effects on caspase-3 activity (Fig 1C). Moreover, CRTH2 expression in cardiomyocytes was gradually increased in response to anoxia in a time-dependent manner (Fig 1D). In a murine model of permanent MI, CRTH2 expression increased significantly in the infarct border zone at day 1 post-MI and peaked at day 3, as compared with the remote region (Fig 1E). These results indicate that $PGD_2$/CRTH2 axis was activated in cardiomyocytes upon anoxic stress.

## CRTH2 inhibition protects against MI by reducing ischemia-induced apoptosis in mice

Anoxia resulted in ~18.2% apoptosis in cultured cardiomyocytes within 1 h; both TUNEL staining (Fig EV1A and B) and flow cytometric analysis (Fig EV1C and D) revealed that CRTH2 deletion significantly decreased anoxia-induced apoptosis in cardiomyocytes, with decreased caspase-3 activity in CRTH2$^{-/-}$ cardiomyocytes (Fig EV1E). In agreement with in vitro observations, CRTH2 deficiency reduced apoptotic TUNEL$^+$ cardiomyocytes in the infarct border zones after MI in mice (Fig 2A and B) and suppressed caspase-3 activity in ischemic hearts (Fig 2C) without affecting necrosis and autophagy (Appendix Fig S2A and B), therefore improving cardiac functions in mice at day 14 after MI (Fig 2D–F). Heart dissection showed significant reduction of infarction size in CRTH2$^{-/-}$ mice compared with WT mice ($28.5 \pm 2.2\%$ vs. $19.7 \pm 1.4\%$, $P < 0.01$; Fig 2G and H). Moreover, CRTH2$^{-/-}$ mice had lower ratio of heart mass to body weight (HW/BW, Fig 2I), higher survival rate (87.5% of CRTH2$^{-/-}$ versus 68.4% of WT; Fig 2J), and less cardiac collagen deposition at day 14 post-MI than WT mice (Fig 2K and L). Similarly, CRTH2 blockade with selective antagonist CAY10595 protected hearts from MI as evidenced by increasing heart functions (Fig EV2A–C) and reducing infarction sizes of hearts (Fig EV2D–E). TUNEL immunostaining also confirmed reduced cardiomyocyte apoptosis in the hearts post-MI in CAY10595-treated mice (Fig EV2F–G). However, no significant differences were found in capillary density (CD31$^+$) (Appendix Fig S3A and B) and mRNA expression of pro-angiogenic growth factors, such as VEGF, FGF, HGF, and PDGF (Appendix Fig S3C), in cardiac tissues from the area at risk at day 14 post-MI between WT and CRTH2$^{-/-}$ mice.

Functional cardiomyocytes can be reprogrammed from fibroblasts by three transcriptional factors—Gata4, Mef2c, and Tbx5 (GMT; Ieda et al, 2010; Wang et al, 2015). We further investigated the effect of CRTH2 deficiency on reprogramming of cardiac fibroblasts into cardiomyocytes using GMT system. Immunostaining revealed that cardiac troponin T (cTnT)-positive cells (~15%) were induced from cardiac fibroblasts by GMT, but no significant difference of reprogramming efficiency was detected between WT and CRTH2$^{-/-}$ fibroblasts (Fig EV3A and B); we also observed similar beating rates of cTnT-positive cells transdifferentiated from WT and CRTH2$^{-/-}$ fibroblasts at different time points tested (Fig EV3C). Consistently, similar expression levels of cardiomyocyte-specific genes were induced in WT and CRTH2$^{-/-}$ fibroblasts by GMT

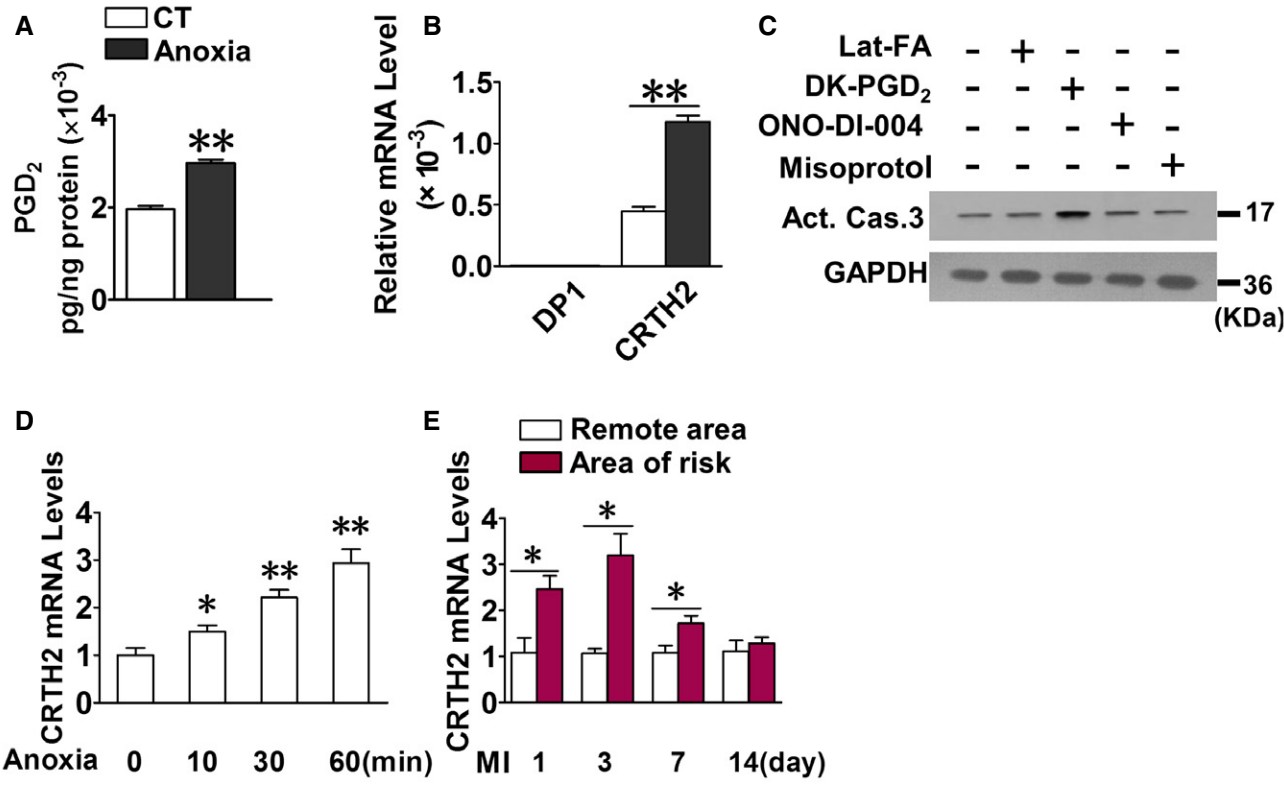

**Figure 1.  PGD$_2$/CRTH2 axis is upregulated in cardiomyocytes in response to anoxia.**

A   The PGD$_2$ production in neonatal mouse cardiomyocytes challenged by anoxia for 1 h. Data represent mean ± SEM.**$P$ < 0.0001 vs. control (Mann–Whitney $U$-test); $n$ = 6.

B   Relative mRNA levels of PGD$_2$ receptors in mouse cardiomyocytes exposed to anoxia. Data represent mean ± SEM. **$P$ < 0.0001 vs. control (Mann–Whitney $U$-test); $n$ = 6.

C   Western blot analysis of caspase-3 in mouse cardiomyocytes treated with PG receptor agonists under anoxia condition. Lat-FA, FP agonist; DK-PGD$_2$, CRTH2 agonist; ONO-DI-004, EP1 agonist; misoprostol, EP4 agonist.

D   Relative mRNA levels of CRTH2 in mouse cardiomyocytes challenged by anoxia in a time-dependent manner. Data represent mean ± SEM. Anoxia 10 min, *$P$ = 0.029, vs. 0 min (one-way ANOVA); anoxia 30 min, **$P$ = 0.000284 vs. 0 min (one-way ANOVA); anoxia 60 min, **$P$ = 0.00016, vs. 0 min (one-way ANOVA); $n$ = 6.

E   Relative mRNA levels of CRTH2 in the different regions of mouse heart post-MI. Data represent mean ± SEM. MI day 1, *$P$ = 0.000765, vs. remote area (unpaired two-tailed $t$-test); MI day 3, *$P$ = 0.0003, vs. remote area (unpaired two-tailed $t$-test); MI day 7, *$P$ = 0.00542, vs. remote area (unpaired two-tailed $t$-test); $n$ = 6.

Source data are available online for this figure.

transduction (Fig EV3D). These results suggested that CRTH2 is not involved in cardiac reprogramming from fibroblast.

## CRTH2$^{+/+}$ bone marrow (BM) reconstitution does not influence cardiac repair after MI in CRTH2$^{-/-}$ mice

An appropriate inflammatory response is required for cardiac recovery after ischemia (Dutta & Nahrendorf, 2015). Both macrophages (CD68$^+$) and neutrophils (Ly6G$^+$) were recruited in the infarcted hearts in mice within one week after the test. CRTH2 deficiency showed no overt effects on the recruitment of macrophages and neutrophils (Appendix Fig S4A–D) and expression of related cytokines (Appendix Fig S4E) in the infarcted hearts. CRTH2 mediates Th2 migration and activation in inflammation (Satoh *et al*, 2006). CRTH2 deficiency retarded T-cell infiltration (CD4$^+$) in peri-infarct zones (Appendix Fig S5A and B) and reduced the expression of Th2 cytokines (IL-4, IL-5, and IL-13) in the inflamed hearts

after MI (Appendix Fig S5C). To further explore whether diminished T-cell recruitment conferred cardioprotection against ischemia-induced apoptosis after MI in CRTH2$^{-/-}$ mice, we reconstituted CRTH2$^{-/-}$ mice with BM from WT mice (Fig EV4A). Decreased T-cell infiltration in the hearts of CRTH2$^{-/-}$ mice was restored by WT BM transplantation (WT→KO), whereas that of WT mice that received CRTH2$^{-/-}$ BM (KO→WT) had significantly lower T cells resident in the infarcted hearts (Fig EV4B and C). However, WT BM transplantation did not increase the attenuated cardiomyocyte apoptosis in CRTH2$^{-/-}$ mice (WT→KO), and CRTH2$^{-/-}$ BM failed to prevent cardiomyocytes from ischemia-induced apoptosis in WT mice (KO→WT) (Fig EV4D and E). Consistently, CRTH2$^{+/+}$ BM transplantation had no overt effects on cardiac recovery after MI in CRTH2$^{-/-}$ mice (Fig EV4F and G). Thus, the cardioprotection of CRTH2 deficiency against ischemia in mice may not be mainly ascribed to BM-derived inflammatory cells, including Th2.

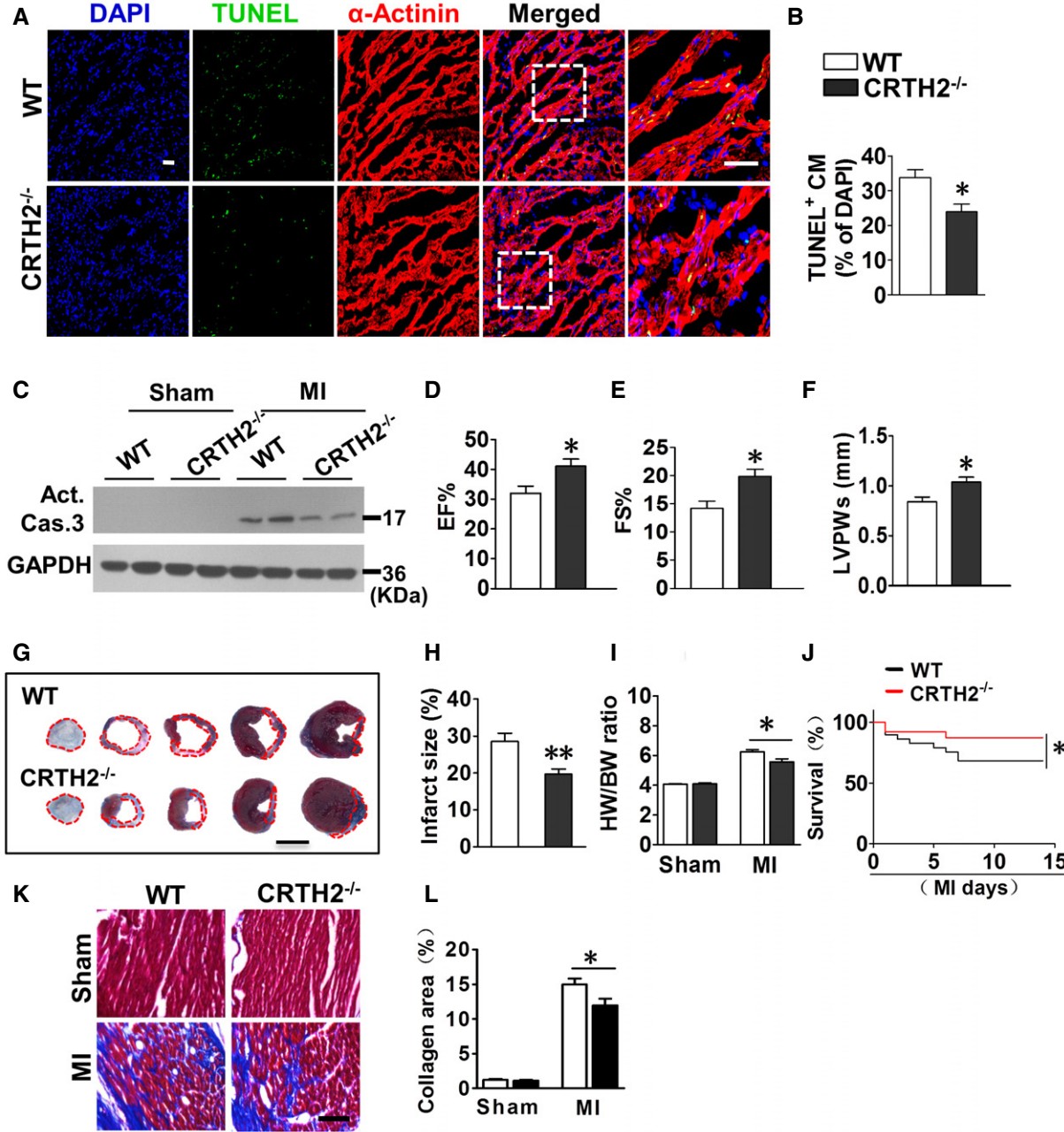

**Figure 2.** CRTH2 inhibition protects against myocardial infarction (MI) by reducing ischemia-induced apoptosis in mice.

A    Representative TUNEL-stained images of the peri-infarct area in MI mouse heart. Green, TUNEL-positive nuclei; blue, DAPI; red, α-actinin; scale bar, 50 μm.
B    Quantification of TUNEL-positive cardiomyocytes. Data represent mean ± SEM. *P = 0.0147, vs. WT (unpaired two-tailed t-test); WT, n = 6; CRTH2$^{−/−}$, n = 8.
C    Western blot analysis of the activated caspase-3 in the infarct border zone of mouse heart post-MI.
D–F  M-mode echocardiographic analysis of cardiac function in mice at day 14 post-MI. EF, ejection fraction (D); FS, fractional shortening (E); LVPWs, left ventricular posterior wall thickness at end-systole. (F). Data represent mean ± SEM. EF, *P = 0.0015, vs. WT (unpaired two-tailed t-test); FS, *P = 0.0015, vs. WT (unpaired two-tailed t-test); LVPWs, *P = 0.00805, vs. WT (unpaired two-tailed t-test); WT, n = 10; CRTH2$^{−/−}$, n = 12.
G    Representative images of Evans blue and TTC-stained mouse heart at day 14 post-MI. Scale bar, 500 μm.
H    Quantification of infarcted size in mouse heart after MI. Data represent mean ± SEM. **P = 0.00354, vs. WT (unpaired two-tailed t-test); n = 10.
I    Heart weight-to-body weight ratio in mice subjected to MI. Data represent mean ± SEM. *P = 0.0119, vs. WT (unpaired two-tailed t-test); WT and CRTH2$^{−/−}$ (Sham), n = 8; WT and CRTH2$^{−/−}$ (MI), n = 10.
J    Kaplan–Meier survival curves for mice at day 14 post-MI. *P = 0.0356, vs. WT (log-rank test).WT, n = 29; CRTH2$^{−/−}$, n = 26.
K    Representative images of Masson's trichrome staining of cardiac tissues from infarcted hearts. Scale bar, 20 μm.
L    Quantification of collagen content in (K). Data represent mean ± SEM. *P = 0.0196, vs. WT (unpaired two-tailed t-test); WT and CRTH2$^{−/−}$ (Sham), n = 7; WT and CRTH2$^{−/−}$ (MI), n = 9.

Source data are available online for this figure.

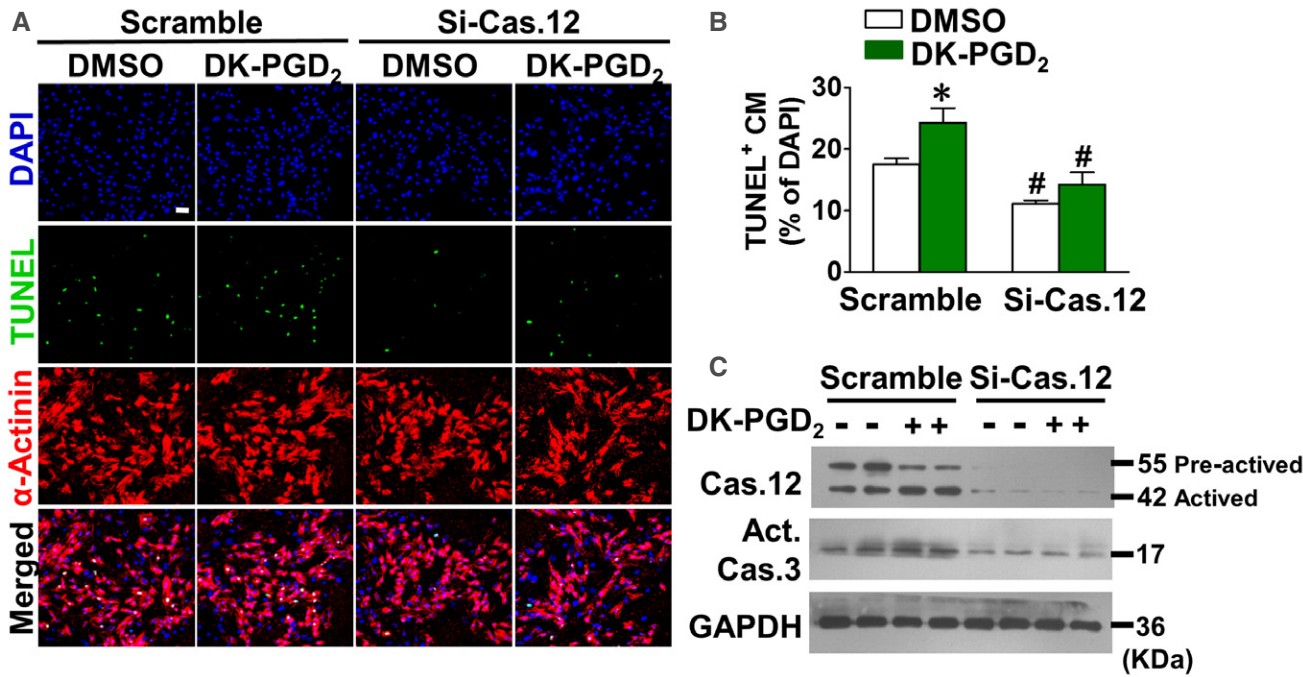

**Figure 3. CRTH2 activation exaggerates anoxia-induced apoptosis in cardiomyocytes through endoplasmic reticulum (ER) caspase-12.**

A   Representative TUNEL-stained images of Si-Cas-12-infected mouse cardiomyocytes after treatment of DK-PGD$_2$ under anoxia condition. Green, TUNEL-positive nuclei; blue, DAPI; red, α-actinin; scale bar, 50 μm.

B   Quantification of TUNEL-positive cardiomyocytes in (A). Data represent mean ± SEM. *$P$ = 0.0222, vs. DMSO + scrambled siRNA; #$P$ = 0.00016, DMSO + si-Cas.12 vs. DMSO+ scrambled siRNA; #$P$ = 0.00695, DK-PGD$_2$ + si-Cas.12 vs. DK-PGD$_2$ + scrambled siRNA (Mann–Whitney *U*-test); $n$ = 7.

C   Western blot analysis of caspase-12 in Si-Cas-12-infected mouse cardiomyocytes after treatment of DK-PGD$_2$ under anoxia condition.

Source data are available online for this figure.

## CRTH2 activation exaggerates anoxia-induced apoptosis in cardiomyocytes through ER caspase-12

Three cardiac apoptotic pathways, such as caspase-8-mediated death receptor pathway, cytochrome C-mediated mitochondrial pathway, and ER caspase-12-initiated pathway, are involved in heart failure (Nakagawa *et al*, 2000; van Empel *et al*, 2005; McIlwain *et al*, 2013). We did not detect marked differences in mitochondrial cytochrome C release (Appendix Fig S6A) and the expression of mitochondrial apoptosis-associated genes (Appendix Fig S6B) in cardiac tissues at risk areas in WT and CRTH2$^{-/-}$ mice. In addition, caspase-8 activity was also not notably changed in infarcted hearts in CRTH2$^{-/-}$ mice compared with control mice (Appendix Fig S6C). However, we observed significant reduction of activated caspase-12 and its downstream substrate-cleaved caspase-9 (activated form) in infarcted hearts of CRTH2$^{-/-}$ mice (Appendix Fig S7A). This result indicates that ER caspase-12-initiated apoptosis was inhibited in ischemic CRTH2$^{-/-}$ hearts. Anoxia induced ER stress in cultured cardiomyocytes as evidenced by the increasing phosphorylation of IRE1 and proteolytic cleavage of ATF6 (Appendix Fig S7B). Moreover, we examined the survival of CRTH2$^{-/-}$ cardiomyocytes in response to non-ER stress [staurosporine (STS) and TNF-α plus cycloheximide (TNF-α/CHX)] and ER stress-inducing apoptotic stimuli (DOX) (Fu *et al*, 2016). CRTH2$^{-/-}$ cardiomyocytes were more resistant to both anoxia and DOX-induced cell death than WT cardiomyocytes

*in vitro*, but displayed similar sensitivity to non-ER stress-inducing apoptotic stimuli (Appendix Fig S7C).

We investigated the role of caspase-12 in anoxia-induced apoptosis in CRTH2 agonist DK-PGD$_2$-treated cardiomyocytes. Caspase-12 knockdown was achieved by adenovirus-guided siRNA approach (Si-Cas.12) (Appendix Fig S7D). DK-PGD$_2$ administration increased anoxia-induced apoptosis in myocytes, which were attenuated by Si-Cas.12 adenovirus infection (Fig 3A and B). Similarly, Si-Cas.12 infection also blunted the augmented caspase-3 activity in DK-PGD$_2$-treated cardiomyocytes, along with diminished caspase-12 expression (Fig 3C). Thus, these results indicate that caspase-12 plays a vital role in CRTH2-mediated cardiomyocyte apoptosis under anoxic stress.

## Treatment with DK-PGD$_2$ impairs cardiac recovery through m-calpain-mediated caspase-12 activation in mice after MI

Accumulated evidence suggested that calpain, a family of Ca$^{2+}$-dependent cytosolic cysteine proteases, is responsible for cleavage and activation of caspase-12 during ER stress-induced apoptosis (Nakagawa & Yuan, 2000; Sanges *et al*, 2006; Martinez *et al*, 2010). Indeed, significantly lower calpain activity was observed in CRTH2$^{-/-}$ cardiomyocytes/hearts than in WT cardiomyocytes/hearts upon anoxia/ischemia (Figs 4A and EV5A). Moreover, the calpain inhibitor—calpeptin—markedly suppressed augmented

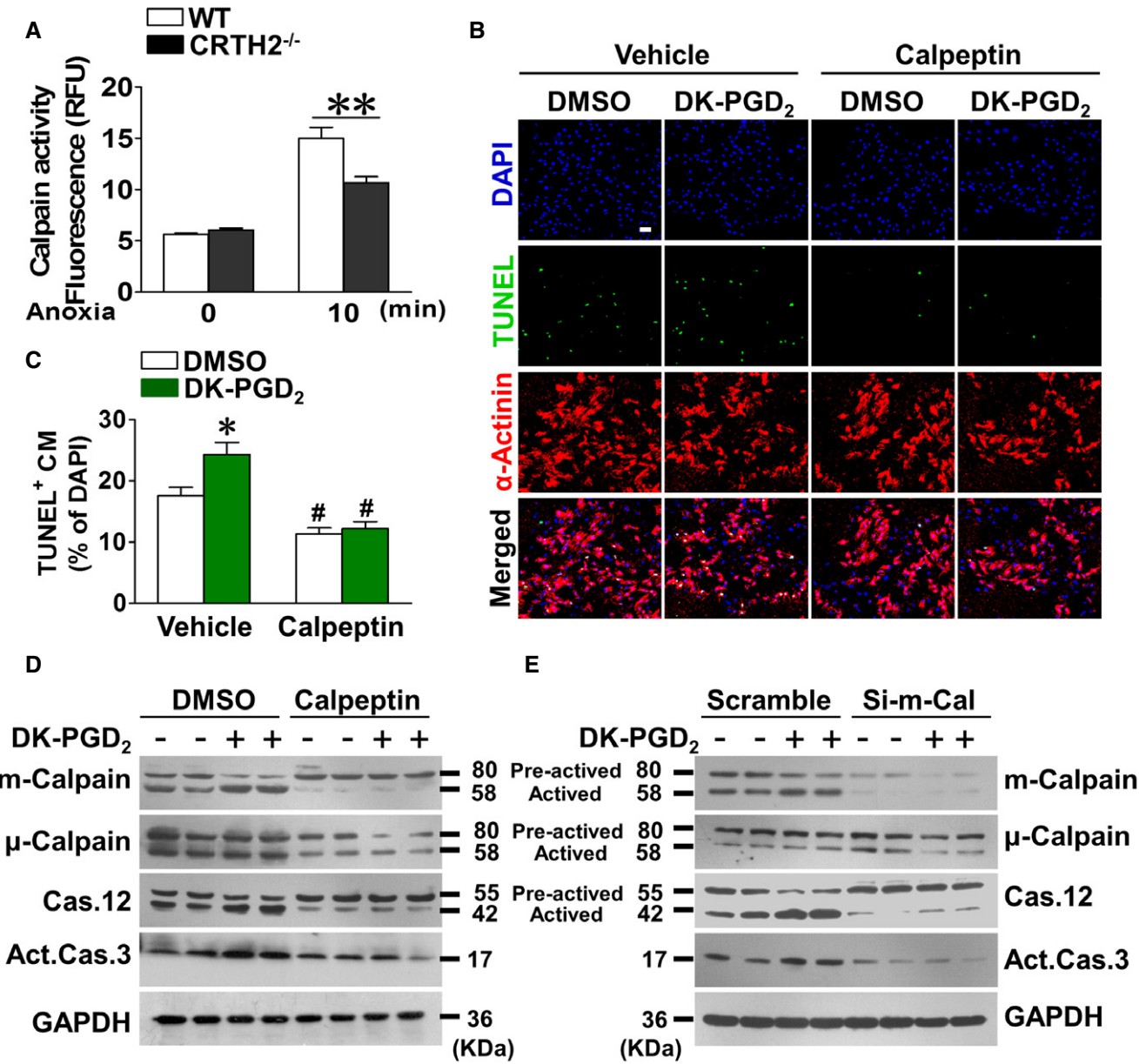

**Figure 4. CRTH2 facilitates caspase-12-mediated apoptosis in cardiomyocytes through m-calpain in response to anoxia.**

A   Calpain activity in mouse cardiomyocytes exposed to anoxia. Data represent mean ± SEM. **$P = 0.00518$, vs. WT (Mann–Whitney $U$-test); anoxia 0 min, WT and CRTH2$^{-/-}$, $n = 4$; anoxia 10 min, WT and CRTH2$^{-/-}$, $n = 6$.

B   Representative TUNEL-stained images of mouse cardiomyocytes treated with calpeptin in the presence of DK-PGD$_2$ under anoxia condition. Green, TUNEL-positive nuclei; blue, DAPI; red, α-actinin; scale bar, 50 μm.

C   Quantification of TUNEL-positive cardiomyocytes in (B). Data represent mean ± SEM. *$P = 0.0136$, vs. DMSO + Vehicle; #$P < 0.0001$, DMSO + Calpeptin vs. DMSO + Vehicle; #$P < 0.0001$, DK-PGD$_2$ + Calpeptin vs. DK-PGD$_2$ + Vehicle (Mann–Whitney $U$-test); DMSO + Vehicle, $n = 9$; DMSO +DK-PGD$_2$, $n = 10$; DMSO + Calpeptin, $n = 10$; DK-PGD$_2$ + Calpeptin, $n = 11$.

D   Western blot analysis of m-calpain, μ-calpain, caspase-12, and caspase-3 in mouse cardiomyocytes treated with calpeptin in the presence of DK-PGD$_2$ under anoxia condition.

E   Western blot analysis of m-calpain, μ-calpain, caspase-12, and caspase-3 in Si-m-Cal-infected mouse cardiomyocytes after treatment of DK-PGD$_2$ under anoxia condition.

Source data are available online for this figure.

apoptosis in CRTH2 agonist DK-PGD$_2$-treated cardiomyocytes upon anoxic challenge (Fig 4B and C) and also decreased the cleavage of caspase-12 and caspase-3 as anticipated (Fig 4D). Three forms of calpains, namely μ-calpain, m-calpain, and calpain-7, were dominantly expressed in cardiomyocytes (Fig EV5B) and ischemic hearts (Fig EV5C). Only m-calpain activity, but not μ-calpain and

calpain-7, was downregulated in the cardiac tissues of risk areas from CRTH2$^{-/-}$ mice (Fig EV5D). In addition, CRTH2 deficiency had no influence on the expression of endogenous calpain inhibitors (Gas2 or calpastatin; Benetti *et al*, 2001; Yang *et al*, 2013b) in mice after MI (Fig EV5E). CRTH2 activation by DK-PGD$_2$ facilitated activation of m-calpain in anoxia-challenged myocytes, without overt effects on μ-calpain and calpain-7 (Fig EV5F and G). Consistently, silence of either μ-calpain or calpain-7 using adenovirus system had no effects on DK-PGD$_2$-triggered caspase-12 activation in cardiomyocytes (Fig EV5F and G). In contrast, m-calpain silence abolished the increased caspase-12 and caspase-3 activities induced by DK-PGD$_2$ in cardiomyocytes under anoxia condition (Fig 4E). In agreement with observations in the culture, calpeptin significantly abrogated the increased infarct sizes (Fig 5A and B) and improved the impaired left ventricular functions in DK-PGD$_2$-treated mice after MI (Fig 5C and D). Histologically, calpeptin markedly reduced the augmented apoptosis (Fig 5E and F) and elevated caspase-3 and caspase-12 activities in cardiac tissues at border zones in DK-PGD$_2$-treated mice (Fig 5G). Hence, CRTH2 activation promoted anoxia-induced apoptosis in myocytes by m-calpain activation.

### CRTH2 couples with G$_{αq}$ to activate m-calpain in cardiomyocytes through intracellular Ca$^{2+}$ mobilization

Calpain activity is mainly dependent on the intracellular calcium (Ca$^{2+}$) status, and CRTH2 can couple G$_{q/11}$ proteins (Schrage *et al*, 2015) to increase intracellular Ca$^{2+}$ level (Hirai *et al*, 2001). We hypothesized that CRTH2 activation could induce calpain activity through triggering intracellular Ca$^{2+}$ release. The mobilization of intracellular Ca$^{2+}$ was significantly compromised in CRTH2$^{-/-}$ cardiomyocytes in response to anoxia (Fig 6A and B). Furthermore, U73122, a phospholipase C inhibitor that blocks calcium entry, significantly attenuated the DK-PGD$_2$-induced cardiomyocyte apoptosis under anoxia condition (Fig 6C and D) by reducing m-calpain and caspase-12 activities (Fig 6E). Co-immunoprecipitation experiments also confirmed that CRTH2 interacted with G$_{q/11}$ (Fig 6F). These results indicate that CRTH2 promotes calpain activity to initiate cardiomyocyte apoptosis under ER stress by triggering G$_{αq}$-dependent Ca$^{2+}$ influx.

### CRTH2 deletion reduces DOX-induced cardiomyocyte apoptosis and cardiac injury in mice

Doxorubicin is an effective chemotherapeutic agent that displays severe cardiotoxicity (Singal & Iliskovic, 1998). As described previously (Fu *et al*, 2016), DOX treatment increased ER stress markers in cardiomyocytes including p-IRE1 and cleaved ATF6 (Fig 7A). Interestingly, DOX significantly upregulated CRTH2 gene expression in cardiomyocytes in a time-dependent manner (Fig 7B). Deletion of CRTH2 markedly ameliorated the left ventricular function and improved the survival rate in mice after DOX treatment (Fig 7C–F) by suppressing DOX-induced cardiomyocyte apoptosis (Fig 7G and H). Importantly, CRTH2 deficiency also led to the reduction of m-calpain, caspase-12, and caspase-3 activities in cardiac tissues from DOX-treated mice (Fig 7I). Taken all together, CRTH2 activation promotes ER stress-induced cardiomyocyte apoptosis through the m-calpain/caspase-12 signaling pathway (Fig 7J).

### Caspase-4 silencing reduces anoxia-induced apoptosis in DK-PGD$_2$-treated human cardiomyocytes

In most humans, caspase-12 appears to be nonfunctional due to a truncating mutation (Fischer *et al*, 2002). Human caspase-4 is the most homologous to mouse caspase-12 and alternatively involved in mediation of ER stress-induced apoptosis in various human cells (Hitomi *et al*, 2004; Li *et al*, 2013; Matsunaga *et al*, 2016; Montagnani Marelli *et al*, 2016). To test whether human caspase-4 mediates CRTH2 activation-induced apoptosis in human cardiomyocytes, we used adenovirus system to silence caspase-4 in AC16 human cardiomyocytes (Fig 8A). As shown in Fig 8B–D, CRTH2 agonist DK-PGD$_2$, indeed, promoted caspase-4 activity in AC16 cells under anoxia condition, along with increased caspase-3 activity. Caspase-4 silence attenuated the increased caspase-3 activity in DK-PGD$_2$-treated AC16 cells, thus suppressing the enhanced apoptosis in AC16 cells treated with DK-PGD$_2$.

## Discussion

Myocardial ischemia is a common stress condition that results in loss of cardiomyocytes. In this study, we observed that both ischemia and DOX treatment upregulated PGD$_2$ receptor CRTH2 expression in cardiomyocytes, along with increasing ER stress. Disruption of CRTH2 receptor markedly improved cardiac function after MI and DOX challenge by attenuating apoptosis in cardiomyocytes. *In vitro*, CRTH2 agonist promoted anoxia-induced apoptosis through activating ER-specific caspase-12. Mechanistically, CRTH2 coupled with G$_{αq}$ activated caspase-12 in cardiomyocytes by Ca$^{2+}$-dependent m-calpain activation. Calpain inhibition or knockdown of caspase-12 blocked CRTH2-augmented apoptosis in anoxia-treated cardiomyocytes. Therefore, CRTH2 activation facilitates ER stress-induced apoptosis through the m-calpain/caspase-12 pathway.

We and others found that massive PGs are generated in the heart after MI (Kong *et al*, 2016; Tang *et al*, 2017) or DOX challenge (Robison & Giri, 1987). Moreover, cardiac-generated PGs, particularly PGD$_2$, mediate cardiac myocyte apoptosis after myocardial ischemia (Qiu *et al*, 2012). PGD$_2$ receptor CRTH2, originally cloned in Th2 cells (Nagata *et al*, 1999), is also highly expressed in cardiac tissues (Sawyer *et al*, 2002), and its expression is also increased in response to ER stress, such as anoxia and DOX treatment. Genetic ablation of CRTH2 ameliorated cardiac myocyte loss and facilitated cardiac recovery after MI and DOX treatment. A previous study reported that COX-2 inhibitor treatment improves left ventricular function and mortality in a murine model of DOX-induced heart failure (Delgado *et al*, 2004). This result suggests that COX-2/PGD$_2$/CRTH2 axis is involved in heart injury induced by DOX, perhaps by ischemia. Interestingly, PGD$_2$ dehydration product, 15-deoxy-$\Delta^{12,14}$-PGJ$_2$, promotes apoptosis in cultured cardiomyocytes through the CRTH2/MAPK/TNF-α-mediated apoptotic pathway (Koyani *et al*, 2014). Paradoxically, investigators also reported that treatment with 15-deoxy-$\Delta^{12,14}$-PGJ$_2$ decreases TNF-α production in cardiomyocytes and reduces myocardial damage of ischemia/reperfusion (Zingarelli *et al*, 2007; Liu *et al*, 2009). The discrepancy may be attributed to the activation of PPAR-gamma by 15-deoxy-$\Delta^{12,14}$-PGJ$_2$ (Harmon *et al*, 2011). In agreement with our observations, PGD$_2$/CRTH2 axis

**Figure 5.  Treatment with DK-PGD₂ impairs cardiac recovery through m-calpain-mediated caspase-12 activation in mice after MI.**

A    Representative images of Evans blue and TTC staining of infarcted mouse heart at day 14 post-MI; scale bar, 500 μm.

B    Quantification of infarcted size in mouse heart after MI. Data represent mean ± SEM. *$P$ = 0.0234, vs. DMSO + Vehicle; [#]$P$ = 0.0276, DMSO + calpeptin vs. DMSO + Vehicle; [#]$P$ = 0.00067, DK-PGD₂ + calpeptin vs. DK-PGD₂ + Vehicle (unpaired two-tailed $t$-test); $n$ = 5.

C, D  M-mode echocardiographic analysis of cardiac function in mice at 14 days after MI. EF, ejection fraction (C); FS, fractional shortening (D); data represent mean ± SEM. EF, *$P$ = 0.0423, vs. DMSO + Vehicle; [#]$P$ = 0.0339, DMSO + calpeptin vs. DMSO + Vehicle; [#]$P$ = 0.00211, DK-PGD₂ + calpeptin vs. DK-PGD₂ + Vehicle (unpaired two-tailed $t$-test); FS, *$P$ = 0.0453, vs. DMSO + Vehicle; [#]$P$ = 0.0259, DMSO + calpeptin vs. DMSO + Vehicle; [#]$P$ = 0.00159, DK-PGD₂ + calpeptin vs. DK-PGD₂ + Vehicle (unpaired two-tailed $t$-test); DMSO + Vehicle, $n$ = 9; DK-PGD₂ + Vehicle, $n$ = 9; DMSO + calpeptin, $n$ = 9; DK-PGD₂ + calpeptin, $n$ = 7.

E    Representative TUNEL-stained images of the peri-infarct area of the heart in mice treated with calpeptin in the presence of DK-PGD₂ at 14 days after MI. Green, TUNEL-positive nuclei; blue, DAPI; red, α-actinin; scale bar, 50 μm.

F    Quantification of TUNEL-positive cardiomyocytes in (E). Data represent mean ± SEM. *$P$ = 0.0479, vs. DMSO + Vehicle; [#]$P$ = 0.000103, DMSO + calpeptin vs. DMSO + Vehicle; [#]$P$ < 0.0001, DK-PGD₂ + calpeptin vs. DK-PGD₂ + Vehicle (unpaired two-tailed $t$-test); DMSO + Vehicle, $n$ = 9, DK-PGD₂ + Vehicle, $n$ = 7, DMSO + calpeptin, $n$ = 10, DK-PGD₂ + calpeptin, $n$ = 10.

G    Western blot analysis of m-calpain, μ-calpain, caspase-12, and caspase-3 in cardiac tissues at border zones of mice treated with calpeptin in the presence of DK-PGD₂ at 24 h after MI.

Source data are available online for this figure.

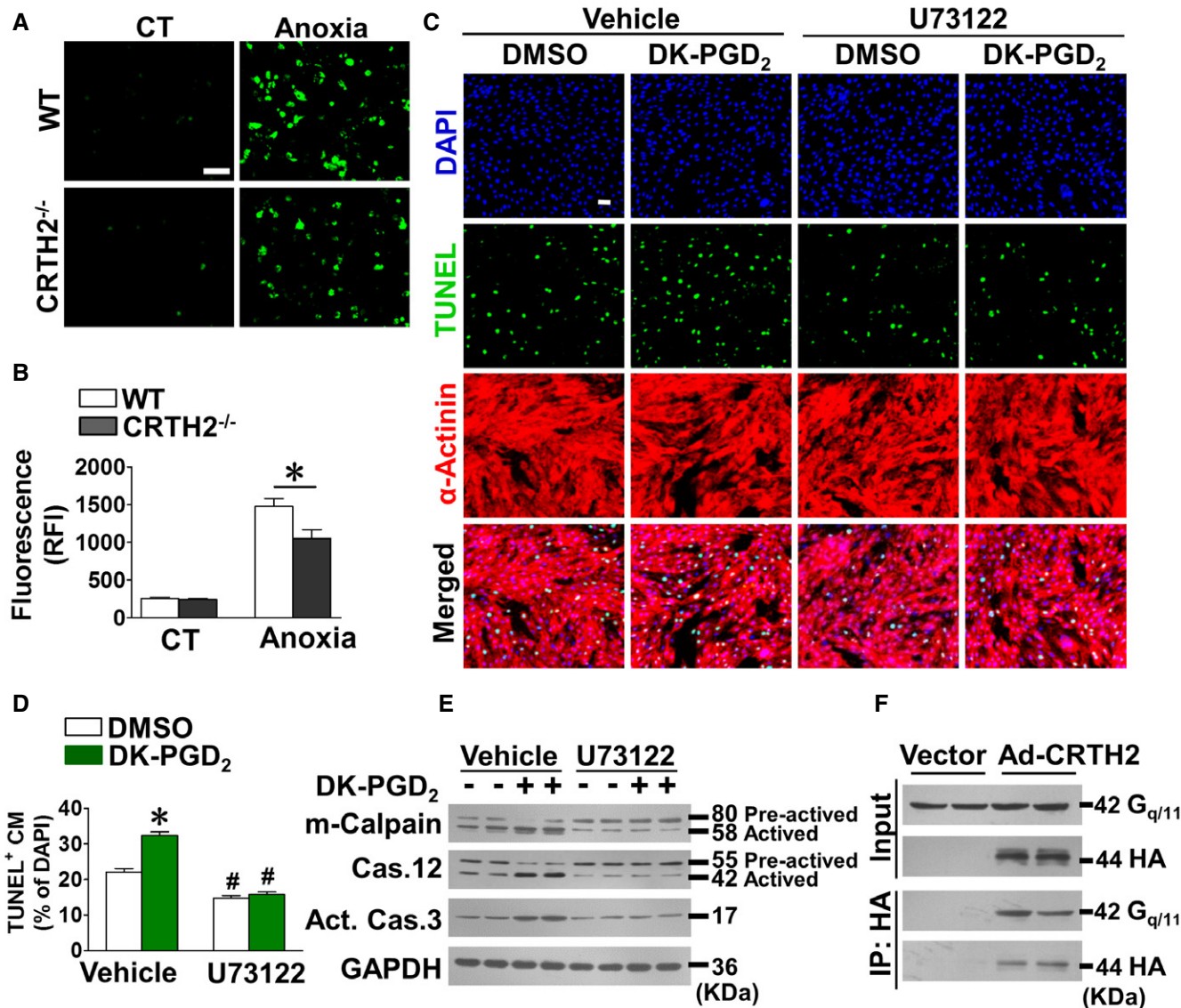

**Figure 6. CRTH2 is coupled to $G_{\alpha q}$ to activate m-calpain in cardiomyocytes through intracellular $Ca^{2+}$ mobilization.**

A   Representative images of Fluo-3 fluorescence (green) in mouse cardiomyocytes challenged with anoxia; Scale bar, 50 μm.

B   Quantification of fluorescence intensities in (A). Data represent mean ± SEM. *P = 0.0151, vs. WT (unpaired two-tailed *t*-test); WT and CRTH2$^{-/-}$ (Control, CT), n = 8; WT and CRTH2$^{-/-}$ (Anoxia), n = 10.

C   Representative TUNEL-stained images of mouse cardiomyocytes treated with U73122 in the presence of DK-PGD$_2$ under anoxia condition. Green, TUNEL-positive nuclei; blue, DAPI; red, α-actinin; scale bar, 50 μm.

D   Quantification of TUNEL-positive cardiomyocytes in (C). Data represent mean ± SEM. *P < 0.0001, vs. DMSO + Vehicle; #P < 0.0001, DMSO + U73122 vs. DMSO + Vehicle; #P < 0.0001, DK-PGD$_2$ + U73122 vs. DK-PGD$_2$ + Vehicle (Mann–Whitney *U*-test); DMSO + Vehicle, n = 9; DK-PGD$_2$ + Vehicle, n = 9; DMSO + U73122, n = 10; DK-PGD$_2$ + U73122, n = 10.

E   Western blot analysis of m-calpain, caspase-12, and caspase-3 in mouse cardiomyocytes treated with U73122 in the presence of DK-PGD$_2$ under anoxia condition.

F   Co-immunoprecipitation of CRTH2 and G$_{q/11}$ in cardiomyocytes from CRTH2$^{-/-}$ mice transfected with HA-tagged CRTH2-expressing adenovirus or control vector.

Source data are available online for this figure.

mediates apoptosis of human osteoclasts by activating caspase-9, not caspase-8 (Yue *et al*, 2012).

Calpains, a family of proteases, are involved in a multitude of physiological and pathological conditions through modulating the functions of their substrates (Ono *et al*, 2016). μ-calpain and m-calpain are the most ubiquitously expressed family members and can be activated *in vitro* by micromolar and millimolar concentrations of calcium, respectively (Ono *et al*, 2016). However, genetic studies have shown mice lacking μ-calpain gene grow normally (Azam *et al*, 2001), but m-calpain-deficient mice are embryonically lethal (Dutt *et al*, 2006), suggesting m-calpain plays more important physiological functions *in vivo* than μ-calpain. In addition to $Ca^{2+}$ binding, phosphorylation of calpain can also increase its activity (Xu & Deng, 2004, 2006),

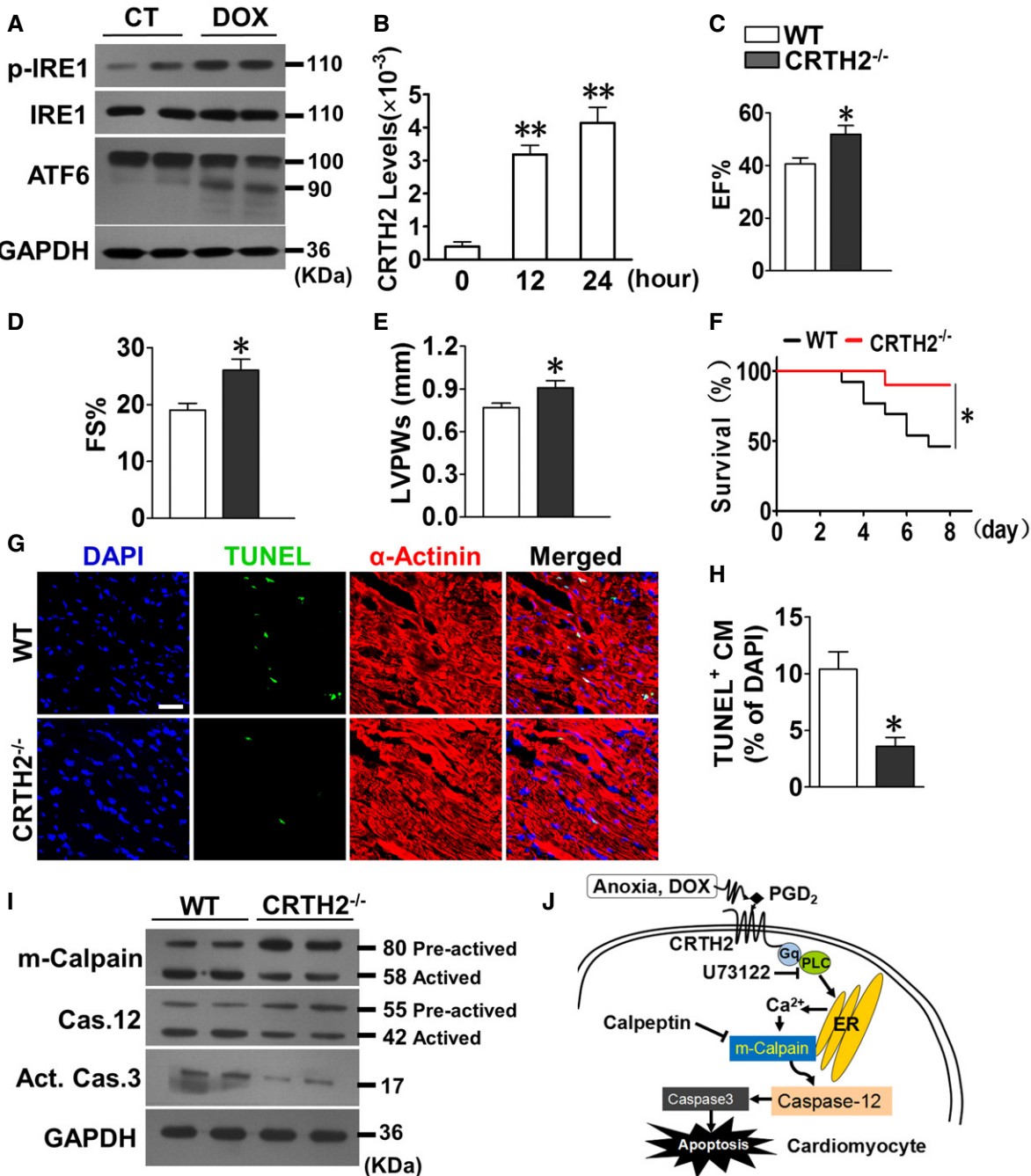

**Figure 7. CRTH2 deletion reduces doxorubicin-induced cardiomyocyte apoptosis and cardiac injury in mice.**

A    Western blot analysis of ER stress markers p-IRE1 and cleaved ATF6 in mouse cardiomyocytes upon DOX (1 μmol/l) treatment.

B    CRTH2 mRNA expression in mouse cardiomyocytes under DOX treatment. Data represent mean ± SEM. **$P$ = 0.00011, 12 h vs. 0 h; **$P$ = 0.000267, 24 h vs. 0 h (one-way ANOVA); $n$ = 4.

C–E    Cardiac function in mice assessed by M-mode echocardiographic analysis on day 7 after DOX treatment (20 mg/kg, i.p.). EF, ejection fraction (C); FS, fractional shortening (D); LVPWs, left ventricle posterior wall thickness at end-systole (E). Data represent mean ± SEM. EF, *$P$ = 0.00836, vs. WT; FS, *$P$ = 0.00383, vs. WT; LVPWs, *$P$ = 0.0198, vs. WT (unpaired two-tailed $t$-test); WT, $n$ = 15, CRTH2$^{-/-}$, $n$ = 13.

F    Kaplan–Meier survival curves for mice subjected to DOX treatment. *$P$ = 0.0305, vs. WT (log-rank test); $n$ = 15.

G    Representative TUNEL-stained images of heart tissue in mice on day 7 after DOX treatment. Green, TUNEL-positive nuclei; blue, DAPI; red, α-actinin; scale bar, 50 μm.

H    Quantification of TUNEL-positive cardiomyocytes in (G). Data represent mean ± SEM. *$P$ = 0.00278, vs. WT (unpaired two-tailed $t$-test); $n$ = 6.

I    Western blot analysis of m-calpain, caspase-12, and caspase-3 in heart tissue from DOX-treated mice.

J    Schematic diagram of CRTH2 promoting cardiomyocyte apoptosis under ER stress through the $G_{\alpha q}$/calpain/caspase-12 signaling pathway.

Source data are available online for this figure.

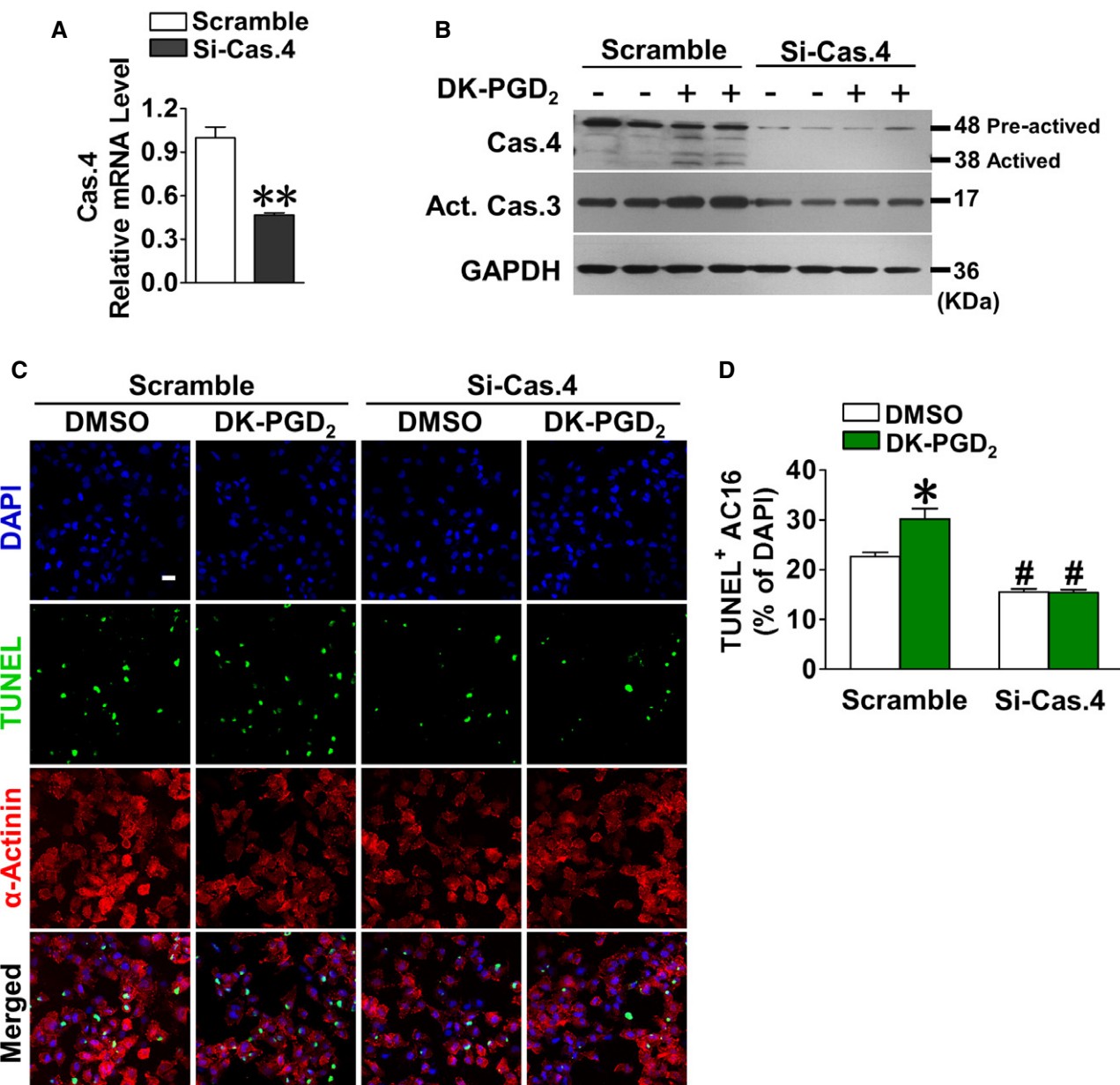

**Figure 8.  Caspase-4 silencing reduces anoxia-induced apoptosis in DK-PGD$_2$-treated human cardiomyocytes.**

A  qRT–PCR depicted the efficiency of adenovirus-mediated siRNA targeting caspase-4 (Si-Cas.4) in human cardiomyocytes (AC16). Data represent mean ± SEM. **$P$ < 0.0001 vs. scrambled siRNA (Mann–Whitney $U$-test); $n$ = 6.

B  Western blot analysis of caspase-4 and caspase-3 in Si-Cas.4-infected AC16 cells after treatment of DK-PGD$_2$ under anoxia condition.

C  Representative TUNEL-stained images of Si-Cas.4-infected AC16 cells after treatment of DK-PGD$_2$ under anoxia condition. Green, TUNEL-positive nuclei; blue, DAPI; red, α-actinin; scale bar, 50 μm.

D  Quantification of TUNEL-positive cardiomyocytes in (C). Data represent mean ± SEM. *$P$ = 0.00723, vs. DMSO + scrambled siRNA; #$P$ < 0.0001, DMSO + si-Cas.4 vs. DMSO + scrambled siRNA; #$P$ < 0.0001, DK-PGD$_2$ + si-Cas.4 vs. DK-PGD$_2$ + scrambled siRNA (Mann–Whitney $U$-test); $n$ = 6.

Source data are available online for this figure.

which, in turn, may influence its sensitivity to Ca$^{2+}$ (Du *et al*, 2017). We observed m-calpain, μ-calpain, and calpain-7 were exclusively expressed in cardiomyocytes, and only m-calpain activity was markedly suppressed in CRTH2$^{-/-}$ cardiomyocytes. Calpain-7 lacks the EF-hand domain and its activity does not depend on Ca$^{2+}$ (Osako *et al*, 2010). Genetic approaches further

confirmed m-calpain is involved in CRTH2-mediated apoptosis in cardiomyocytes.

Procaspase-12, which is predominantly located in the ER, can be activated through ER stress (Szegezdi *et al*, 2003) or cleavage by m-calpain at T132 and K158 (Nakagawa & Yuan, 2000). Notably, a significantly lower activity of m-calpain was

detected in CRTH2$^{-/-}$ cardiomyocytes. Inhibition of m-calpain effectively protected cardiomyocytes against CRTH2 agonist-induced apoptosis under ER stress by suppressing caspase-12 activity. Similarly, calpain-mediated caspase-12 activation is also involved in TNF-$\alpha$-induced apoptosis in cardiomyocytes (Bajaj & Sharma, 2006), neuron apoptosis in retinitis pigmentosa (Sanges *et al*, 2006), and hydroxyeicosatetraenoic acid-induced fibroblast apoptosis (Nieves & Moreno, 2007). In humans, functional caspase-12 is lost due to a frameshift mutation and a premature stop codon in its transcript (Fischer *et al*, 2002). Caspase-4, homolog to mouse caspase-12, can function as ER stress-induced caspase in human cells (Hitomi *et al*, 2004; Simard *et al*, 2015; Montagnani Marelli *et al*, 2016). Knocking down caspase-4 attenuated anoxia-induced apoptosis in CRTH2 agonist-treated human cardiomyocytes. Calpain activity is mainly dependent on the intracellular $Ca^{2+}$ (Potz *et al*, 2016). We observed that CRTH2 was coupled with $G_{\alpha q}$ in cardiomyocytes. CRTH2 deficiency markedly reduced intracellular $Ca^{2+}$ levels in cardiomyocytes in response to anoxia. Blockage of calcium entry dramatically suppressed m-calpain/caspase-12 activity and subsequently reduced the exaggerated apoptosis in CRTH2 agonist-treated cardiomyocytes. This result indicates that CRTH2-mediated intracellular $Ca^{2+}$ mobilization activates m-calpain/caspase-12 signaling. Accordingly, enhanced $G_{\alpha q}$ signaling has been shown to trigger apoptotic cardiomyopathy and heart failure (Adams *et al*, 1998; Yussman *et al*, 2002). Thus, activation of CRTH2 may facilitate calpain-initiated cardiomyocyte apoptosis under ER stress via the $G_{\alpha q}$-$Ca^{2+}$ pathway.

In summary, stimulation of CRTH2 receptor exerts pro-apoptotic effect in cardiomyocytes via the calpain/caspase-12 signaling pathway. Thus, the inhibition of CRTH2 may have therapeutic potential for apoptotic cardiomyopathy.

# Materials and Methods

### Animals

CRTH2 knockout mice (Satoh *et al*, 2006) and wild-type littermates (C57BL/6J genetic background) were housed in polypropylene cages and maintained in a temperature-controlled (22 ± 1°C) and relative humidity (50 ± 5%) environment with 12-h dark–light cycle and feed by sterile food and water *ad libitum*. Essential cleanliness and sterile condition were adopted according to SPF facilities. All animal experiments were performed in accordance with the approval of the Institutional Animal Care and Use Committee of the Institute for Nutritional Sciences, University of Chinese Academy of Sciences.

### Reagents

Calpeptin, arachidonic acid, CAY10595, and 13, 14-dihydro-15-keto PGD$_2$ were purchased from Cayman Chemical Company (Cayman Chemical, Ann Arbor, MI, USA). U-73122, staurosporine, cycloheximide, and DOX were obtained from Sigma Company (Sigma-Aldrich, St. Louis, MO, USA). TNF-$\alpha$ was purchased from Peprotech (Peprotech Inc., Rocky Hill, NJ, USA).

### Mouse MI model

Mice were subjected to a permanent ligation of the left anterior descending artery or to a sham operation as described previously (Gao *et al*, 2010). Briefly, 6- to 8-week-old male mice were anesthetized with isoflurane and placed on a heating pad (37°C). Heart was pushed out at the fourth intercostal space after a small hole was made with a mosquito clamp. The left anterior descending coronary artery (LAD) was then located, sutured, and ligated at a site ~3 mm from its origin using a 6-0 silk suture. Sham-operated animals were subjected to the same procedures without ligation. Mice were sacrificed at specified days, and hearts were removed and fixed, or dissected for protein or RNA assay. The infarcted size was determined as previously described (Qian *et al*, 2011). The infarct size weight was calculated using the following formula: [(A1 × W1) + (A2 × W2) + (A3 × W3) + (A4 × W4) + (A5 × W5)]. A is the percent of infarcted area in the slice, and W is the weight of the respective slices.

Drug administration in mice was performed as previously described (Mani *et al*, 2009; Maicas *et al*, 2012; Zhang *et al*, 2016). In brief, 6- to 8-week-old male mice were given subcutaneous injections of DK-PGD$_2$ (0.6 mg/kg) and calpeptin (0.5 mg/kg), or CAY10595 (5 mg/kg) 15 min before MI induction. Subsequently, mice were treated with DK-PGD$_2$ (0.3 mg/kg/twice a day) and calpeptin (0.5 mg/kg/day) by subcutaneous injection, or CAY10595 (5 mg/kg/day) by oral gavage until terminal echocardiography procedures were completed. DOX (20 mg/kg) was administered by a single intraperitoneal injection (i.p.), and cardiac function was then assessed by echocardiography 7 days later.

### Isolation and culture of neonatal mouse cardiomyocyte

Neonatal mouse cardiomyocytes were isolated and collected according to the instructions of Neonatal Rat/Mouse Cardiomyocyte Isolation Kit (Cellutron Life Technologies, MD, USA). In brief, hearts from postnatal mice (P1) were separated, and ventricular cells were released with digestion buffer at 37°C.

Ventricular cardiomyocytes were grown in NS medium (Cellutron Life Technology) supplemented with 10% fetal bovine serum. For anoxia culture, cardiomyocytes were placed in an anoxic chamber with a water-saturated atmosphere composed of 5% CO$_2$ and 95% N$_2$ at 37°C as previously described (Mehrhof *et al*, 2001).

### PG extraction and analysis

Cultured cardiomyocytes were incubated with arachidonic acid (30 μmol/l), and culture supernatants were used for PG extraction. After centrifugation at 12,000 *g* for 15 min at 4°C, internal standards (2 μl), 40 μl of citric (1 M), and 5 μl of BHT were added to the sample and then strenuously vibrated with 1 ml of solvent (normal hexane: ethyl acetate, 1:1) for 1 min. After centrifugation at 6,000 *g* for 10 min, the supernatant organic phase was collected and dried under a gentle stream of nitrogen and dissolved in 100 μl of 10% acetonitrile in water. The prostanoid metabolites were quantitated using liquid chromatography/mass spectrometry/mass spectrometry

analyses. The level of PGs was normalized to total protein concentration.

### Cell sorting from post-MI hearts

At indicated time point after MI, male mice were anesthetized and perfused intracardially with 30 ml of pre-cooled phosphate-buffered saline (PBS) to exclude blood cells. Subsequently, the border zone and infarcted region of the heart were dissected, minced, and enzymatically digested with a cocktail of collagenase II (450 U/ml), collagenase XI (125 U/ml), DNase I (60 U/ml), and hyaluronidase (60 U/ml) (Sigma-Aldrich) at 37°C for 1.5 h with gentle shake. The digestion mixture was then passed through a 70-μm cell strainer. Leukocyte-enriched fractions were isolated by density gradient centrifugation on a 40–70% Percoll gradient (GE Healthcare, Uppsala, Sweden). Cells at the 40/70 interface were then collected for inflammatory cytokine gene detection as previously described (Zouggari *et al*, 2013).

### Evaluation of apoptosis

Apoptotic cells in both tissue sections and cultured cells were assayed by the terminal deoxynucleotidyl transferased UTP nick-end labeling (TUNEL) method. TUNEL was performed according to the protocol provided by the manufacturer (Roche Applied Science, Indianapolis IN, USA). Nuclear density was determined by counting DAPI-stained nuclei in 20 different fields for each sample.

### Immunofluorescence staining

Frozen sections and cell climbing slices were fixed in cold acetone, washed with PBS, and then incubated with 3% BSA in PBS for 60 min to block nonspecific binding of the antibodies. Thereafter, the samples were incubated with primary antibodies specific to mouse CD68 (diluted 1:200; Serotec), Ly6G (diluted 1:50; BD Biosciences), CD4 (diluted 1:200; eBioscience), CD31 (diluted 1:500; BD Biosciences), cardiac troponin T (diluted 1:100; Santa Cruz Biotechnology), or α-actinin (diluted 1:1,000; Sigma-Aldrich) overnight at 4°C. Afterward, the slides were washed with PBS three times and incubated with corresponding secondary antibodies conjugated with Alexa Fluor 488 or Alexa Fluor 555 (diluted 1:1000; Invitrogen, Carlsbad, CA, USA) at room temperature for 2 h. Sample was mounted in ProLong Gold antifade reagent with DAPI (Invitrogen). All of the immunofluorescence images were captured and analyzed using a laser-scanning confocal microscope (Carl Zeiss, Oberkochen, Germany). For the MI model, at least 10 random images from three different sections were obtained from each animal. For cells growing on the glass slide, at least five random fields were taken in the central region of each sample (Shen *et al*, 2016).

### Echocardiography

Echocardiography was performed using a Visual Sonics Vevo 770 high-resolution imaging system with a 15-MHz linear-array transducer (Visual Sonics Inc., Toronto, Canada). Male mice were anesthetized with isoflurane, and the heart was imaged in the two-dimensional parasternal short-axis view. An M-mode echocardiogram of the midventricle was then recorded at the level of papillary muscles. Cardiac function was evaluated as previously described (Liao *et al*, 2002).

### Bone marrow transplantation

Bone marrow transplantation (BMT) was performed as previously described (Shi *et al*, 2014). In brief, male mice (6–8 weeks old) were euthanized, and BM cells were collected from the femurs and tibias. Recipient mice were lethally irradiated with a total of 9.5 Gy of total body irradiation administered in three bursts (one 3.5-Gy dose and two 3-Gy doses administered 1.5 h apart) from a $^{137}$Cs source (MDS Nordion, Ottawa, Ontario, Canada) and transplanted with $5 \times 10^{6}$ donor BM cells via tail vein injection to reconstitute the hematopoietic system. Eight weeks after transplantation, BMT chimeric mice were used for experiments.

### Western blot analysis

Protein from the heart and cardiomyocytes was extracted in the lysis buffer with protease inhibitors. The protein concentrations were determined using Pierce BCA Protein Assay Kit (Pierce, Rockford, IL, USA). Equivalent levels of proteins were denatured and resolved with 10% sodium dodecyl sulfate–polyacrylamide gel electrophoresis gels and then transferred to nitrocellulose membranes, incubated with 5% skimmed milk, and probed with primary antibodies overnight at 4°C. Primary antibodies were diluted as follows: anti-cleaved-caspase-3 (1:1,000; Signalway Antibody LLC), anti-mouse GAPDH, anti-HA-tag, anti-m-calpain, anti-μ-calpain, anti-caspase-12, anti-IRE1 (1:1,000; Cell Signaling Technology), anti-calpain-7 (1:1,000; ProteinTech Group, Chicago, IL, USA), anti-P-IRE1(1:1,000; Littleton, CO, USA), anti-ATF6 (1:500, Santa Cruz, CA, USA), anti-LC3A/B, anti-phospho-MLKL, anti-MLKL (1:1,000; Cell Signaling Technology), anti-caspase-4, anti-caspase-8, and anti-caspase-9 (1:1,000; ABclonal). The membranes were washed and then incubated in horseradish peroxidase-labeled secondary antibody for 1–2 h at room temperature. Proteins were detected using enhanced chemiluminescence reagents (Thermo Scientific, Waltham, MA, USA), and blots were quantified with ImageJ and normalized by GAPDH.

### Adenovirus generation and infection

Adenoviruses were constructed using the AdEasy Adenoviral System (Qibogene, Irvine, CA) as previously described (Luo *et al*, 2007). In brief, the pAd-Track-CMV-GFP (+) vector containing full-length cDNA that encodes mouse CRTH2 or the pAd-Track-CMV-GFP (−) vector containing specific siRNA hairpin was generated in the HEK293 viral packaging cell line. After several rounds of amplification, recombinant adenovirus was purified by ultracentrifugation in a density cesium chloride gradient. The infection efficiency was estimated by determining the fluorescence of GFP. Adenoviral infection of cardiomyocytes was performed as described previously (Sundaresan

*et al*, 2012). All the siRNA sequences are listed in Appendix Table S1.

### RNA extraction and quantitative real-time polymerase chain reaction

Total RNA from the hearts or cardiomyocytes was extracted using TRIzol reagent (Invitrogen) according to the manufacturer's instructions. Total RNA (1 μg) was reverse-transcribed to cDNA using a Reverse Transcription Reagent kit (Takara, Dalian, China) according to the manufacturer's method. Target gene expression was normalized to the level of *gapdh* mRNA. The quantitative real-time polymerase chain reaction (qRT–PCR) protocol was as follows: 5 min at 95°C for one cycle, followed by 40 cycles at 95°C for 30 s, 60°C for 30 s, and 72°C for 25 s, and a final extension at 72°C for 10 min. Dissociation curve was obtained for each PCR product. All the primer sequences are listed in Appendix Table S2.

### CCK-8 assay

Cardiomyocytes were seeded into 96-well plates at the same density and cultured for 8 h. At the end of the culture, the medium was removed and replaced with 100 μl of fresh medium and 10 μl of Cell Counting Kit-8 (CCK-8) assay kit reagent (Dojindo Laboratories, Kumamoto, Japan). Plates were incubated at 37°C for 1 h, and the absorbance was then measured at 450 nm using a SpectraMax 190 microplate reader (Molecular Devices). Background absorbance value (from wells without cells) was subtracted from all values.

### Calpain activity assay

Calpain activity assay was performed using a kit according to the manufacturer's instructions (Calbiochem, San Diego, CA, USA). Fluorescence value was recorded at an excitation wavelength of 360–380 nm and an emission wavelength of 440–460 nm by using a fluorescence plate reader. Relative fluorescence units were then calculated.

### Confocal calcium imaging

Calcium imaging was recorded in cardiomyocytes using a laser-scanning confocal microscope (Carl Zeiss, Inc., Germany) as previously described (Shen *et al*, 2016). Briefly, cells were treated with 5 μg/ml Fluo-3 (Dojindo Laboratories, Kumamoto, Japan) in Hank's balanced salt solution (HBSS; Invitrogen) at 37°C for 30 min. After washing, cells were incubated in HBSS during calcium imaging. Images were obtained in the line-scan mode with 512 pixels per line at a rate of 5 ms per scan and excited at 488 nm. Two-dimensional images were acquired with the confocal microscope operated at the frame-scan mode (X-Y, 512 × 512 pixels).

### Immunoprecipitation

Cardiomyocytes were transfected with adenovirus harboring CRTH2 cDNA or empty vector with HA-tag. The whole-cell lysates were incubated with 5 μl of HA-tag antibody or normal IgG (Cell Signaling Technology) control at 4°C for 3 h and then incubated with protein A/G agarose (Invitrogen) at 4°C overnight with gentle agitation. After washing three times, the immune complexes were recovered by boiling in SDS loading buffer and then subjected to Western blot analysis with HA-tag antibody or anti-$G_{q/11}$ antibody (Santa Cruz Biotechnology, Santa Cruz, CA, USA).

### Isolation of cardiac fibroblasts

Cardiac fibroblast isolation was performed as described previously (Qian *et al*, 2013; Lalit *et al*, 2016). Briefly, mouse cardiac fibroblasts (CFs) were derived from day 1 to day 3 neonatal pups. Isolated neonatal hearts were minced into small pieces less than 1 mm$^3$ in size, explants were then plated on 0.1% gelatin-coated dishes in fibroblast medium (IMDM/20% FBS) for 7 days. Then the migrated fibroblasts were trypsinized and filtered through 40-μm cell strainers (Thermo Scientific) to remove tissue fragments. Cardiomyocyte contamination was examined before experiments by staining cardiac troponin T (cTnT).

### Production of retroviruses and induction of reprogramming

Cardiac reprogramming of murine cardiac fibroblasts was performed as previously described (Wang *et al*, 2015). Briefly, pMXs-based retroviral vectors (Gata4, Mef2c, Tbx5) were introduced into Plat-E cells using Lipofectamine 2000 transfection reagent (Life Technologies) according to the manufacturer's recommendations. Medium was changed the next day and virus-containing supernatant was collected 48 h after transfection and centrifuged at 48,400 *g* for 2 h at 4°C. Viruses were then re-suspended by fibroblast media supplemented with 4 μg/ml polybrane (Life Technologies) and added to target cells immediately; 24 h after infection, the culture medium was replaced with cardiomyocyte culture medium (iCM medium, 10% FBS of DMEM/M199 (4:1)) and changed every 3–4 days.

### Statistical analysis

All data are expressed as the mean ± standard error of the mean (SEM). Statistical analysis was performed using SPSS version 16.0 software (SPSS Inc., Chicago, IL, USA). Normality of distribution was assessed using the Kolmogorov–Smirnov test. Unpaired Student's *t*-test, Mann–Whitney *U*-test, or one-way analysis of variance followed by Bonferroni *post hoc* test was used to compare two and multiple groups, respectively. Survival rates were compared using the log-rank test. *P*-values of < 0.05 were considered statistically significant.

Expanded View for this article is available online.

### Acknowledgments

We thank Dr. Masataka Nakamura (Tokyo Medical and Dental University) for providing CRTH2-deficient mice. This work was supported by National key R&D Program of China (2017YFC1307404) and the National Natural Science Foundation of China (NSFC) (81790623, 81525004, 91439204, 81030004, 31300944, 81400321, 81400239, 81771513, 31200860, and 31771269), Y. Yu is a Fellow at the Jiangsu Collaborative Innovation Center for Cardiovascular Disease Translational Medicine.

## The paper explained

### Problem

Chemoattractant receptor-homologous molecule expressed on T helper type 2 cells (CRTH2), which mediates recruitment and activation of Th2 cell, is highly expressed in the heart. However, its specific role in ischemic cardiomyopathy is not fully understood.

### Results

We found that activation of CRTH2 promoted ER stress-induced cardiomyocyte apoptosis in mice postmyocardial infarction and doxorubicin challenge through the $G_{\alpha q}$/m-calpain/caspase-12 signaling pathway. CRHT2 activation also induced apoptosis in human cardiomyocytes in response to anoxia by increasing caspase-4 activity, an alternative to caspase-12 in humans.

### Impact

Our result suggested that CRTH2 inhibition may have therapeutic potential for ischemic cardiomyopathy.

## Author contributions

SZ and YY designed research; SZ, YS, DK, CW, JL, YW, QW, SY, JZ, JT, QZ, and LL conducted experiments; SZ, XL, YS, and YY analyzed the data; LQ and ZS provided experimental reagents; SZ, YS, and YY wrote the paper.

## Conflict of interest

The authors declare that they have no conflict of interest.

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
