## [Review Process File · EMBO Molecular Medicine]

CRTH2 promotes endoplasmic reticulum stress-induced cardiomyocyte apoptosis through m-calpain

Shengkai Zuo, Deping Kong, Chenyao Wang, Jiao Liu, Yuanyang Wang, Qiangyou Wan, Shuai Yan, Jian Zhang, Juan Tang, Qianqian Zhang, Luheng Lyu., Xin Li, Zhixin Shan, Li Qian, Yujun Shen, Ying Yu

Review timeline:

Submission date:	06 July 2017
Editorial Decision:	17 August 2017
Revision received:	09 November 2017
Editorial Decision:	30 November 2017
Revision received:	07 December 2017
Accepted:	14 December 2017

Editors Roberto Buccione and Céline Carret

Transaction Report:

1st Editorial Decision

17 August 2017

Thank you for the submission of your manuscript to EMBO Molecular Medicine. I apologise for the delay in reaching a decision on your manuscript. We experienced significant difficulties in securing expert and willing reviewers. Further to this, reviewer #2 unexpectedly made him/herself unavailable.

I am thus proceeding based on the two consistent evaluations obtained so far as further delays cannot be justified.

You will see that both Reviewers are supportive of your work and underline its potential interest. Reviewer 1 raises a few points that require your intervention mostly centred on providing more stringent and convincing experimental support for the observed apoptosis. S/he also lists a few other items that require your action. Reviewer #3 is also positive but notes that to strengthen the clinical relevance of the study, it would be important to verify whether the inhibition of CRTH2 reduces cardiac fibrosis. The reviewer also asks whether CRTH2 deletion is associated to cardiac fibroblast reprogramming. I find that the latter requests are well-taken and important.

In conclusion, while publication of the paper cannot be considered at this stage, we would be pleased to consider a suitably revised submission, provided, however, that all the Reviewers' concerns are fully addressed with further experimentation where required.

***** Reviewer's comments *****

Referee #1 (Remarks):

In this manuscript, Zuo et al present experimental data to suggest that PGD2/CRTH2 axis, an immune regulator, plays an unexpected role mediating cardiomyocyte apoptosis. PGD2 has two membrane receptors, DP1 and CRTH2, but the authors found that only CRTH2 is expressed in

mouse cardiomyocytes, and that PGD2/CRTH2 axis is significantly activated in cardiomyocytes in response to anoxia and doxorubicin treatment, which elevates endoplasmic reticulum stress. Using *in vivo* myocardial infarction and doxorubicin models, the authors show that disruption of CRTH2 improves cardiac recovery and enhanced survival rate by suppressing cardiomyocyte apoptosis. Mechanistically, activation of CRTH2 specifically promoted caspase-12 activity, a key apoptotic proteases in ER stress, in cardiomyocytes in response to anoxia through m-calpain dependent pathway. CRTH2 is coupled with Gαq to elicit intracellular Ca²⁺ flux and subsequently stimulated m-calpain to activate caspase-12 in cardiomyocytes under ER stress. Therefore, CRTH2 mediated ER stress induced-cardiomyocyte apoptosis through the Gαq/m-calpain/caspase-12 signaling pathway. Overall, this is a study of clinically significance. The manuscript is well-written. Data are well presented to support the conclusion. I have only following minor comments for the authors to consider:

1. The authors found that activation of CRTH2 specifically promotes caspase-12 activity; however, initiation of ER stress-induced apoptosis involves transcriptional activation of the C/EBP-homologous protein (CHOP). Does CRTH2 deletion affect CHOP expression?
2. Additional evidence, such as apoptotic DNA ladder or annexin V, would be needed to support the TUNEL data presented in Figure 2.
3. The authors would need to evaluate the Calpain activity and isoform expression pattern at MI border area.
4. The authors would need to discuss the differences between μ-calpain, m-calpain and calpain 7.

Referee #3 (Comments on Novelty/Model System):

I found the study well performed and the experimental approach was appropriate to address the hypothesis proposed. The data reported on the involvement CRTH2 activation in cardiomyocyte apoptosis are convincing. However, it would be important to verify whether CRTH2 deletion or blockage may inhibit cardiac fibrosis

Referee #3 (Remarks):

In the present manuscript, Zuo et al. aimed to study the role of (CRTH2) in cardiac recovery in mice post myocardial infarction and doxorubicin challenge. In particular, they examined the impact of CRTH2 in cardiomyocyte apoptosis. The experimental strategy was to study cardiomyocytes in response to anoxia and DOX treatment *in vitro*; then they assessed the influence of CRTH2 deletion on Mouse MI model. They found that CRTH2 inhibition protects against MI by reducing ischemia-induced apoptosis in mice. CRTH2 activation promoted anoxia-induced apoptosis in myocytes by m-calpain activation. CRTH2 activation promoted ER stress-induced cardiomyocyte apoptosis through the m-calpain/caspase-12 signaling pathway. In AC16 human cardiomyocytes under anoxia condition, CRTH2 activation promoted caspase-4 activity (caspase-4 the most homologous to mouse caspase-12) along with increased caspase-3 activity. Caspase-4 silence attenuated the increased caspase-3 activity in DK-PGD2-treated AC16 cells. Altogether the findings of this study suggest that the stimulation of CRTH2 receptor exerts a pro-apoptotic effect in cardiomyocytes via the calpain/caspase-12 signaling pathway. They propose that the inhibition of CRTH2 may have therapeutic potential for apoptotic cardiomyopathy.

This is a clinically relevant study using appropriate experimental technologies to address the role of CRTH2 in progressive cardiomyocyte loss associated with ischemic heart disease and chemotherapy-induced cardiomyopathy. Interestingly, the results of this study suggest the possible use of selective antagonists of CRTH2 to protect hearts from the consequences of MI. The manuscript is well and clearly written.

Major points

In response to MI, different reparative responses are induced and importantly the fibrotic responses in the damaged heart. Thus, it would be important to know whether, in addition to an anti-apoptotic effect on cardiomyocytes, CRTH2 deletion or blockage might inhibit cardiac fibrosis. Interestingly, CRTH2 is also expressed in fibroblasts.

If possible, it would be very interesting to know whether the CRTH2 deletion in cardiac fibroblasts may be associated to reprogramming of fibroblasts into cardiomyocytes.

1st Revision - authors' response

09 November 2017

Referee #1 (Remarks):

In this manuscript, Zuo et al present experimental data to suggest that PGD2/CRTH2 axis, an immune regulator, plays an unexpected role mediating cardiomyocyte apoptosis. PGD2 has two membrane receptors, DP1 and CRTH2, but the authors found that only CRTH2 is expressed in mouse cardiomyocytes, and that PGD2/CRTH2 axis is significantly activated in cardiomyocytes in response to anoxia and doxorubicin treatment, which elevates endoplasmic reticulum stress. Using in vivo myocardial infarction and doxorubicin models, the authors show that disruption of CRTH2 improves cardiac recovery and enhanced survival rate by suppressing cardiomyocyte apoptosis. Mechanistically, activation of CRTH2 specifically promoted caspase-12 activity, a key apoptotic proteases in ER stress, in cardiomyocytes in response to anoxia through m-calpain dependent pathway. CRTH2 is coupled with Gαq to elicit intracellular Ca²⁺ flux and subsequently stimulated m-calpain to activate caspase-12 in cardiomyocytes under ER stress. Therefore, CRTH2 mediated ER stress induced-cardiomyocyte apoptosis through the Gαq/m-calpain/caspase-12 signaling pathway. Overall, this is a study of clinically significance. The manuscript is well-written. Data are well presented to support the conclusion. I have only following minor comments for the authors to consider:

1. The authors found that activation of CRTH2 specifically promotes caspase-12 activity; however, initiation of ER stress-induced apoptosis involves transcriptional activation of the C/EBP-homologous protein (CHOP). Does CRTH2 deletion affect CHOP expression?

Thanks for the insightful comments. CCAAT/enhancer-binding protein homologous protein (CHOP) is an endoplasmic reticulum stress-inducible protein that plays a critical role in the regulation of programmed cell death [*Nat Rev Mol Cell Biol.* 2007, 8: 519–529]. Indeed, CHOP is involved in ER stress-mediated apoptosis in cardiomyocytes such as pressure overload and ischemia/reperfusion [*Circulation.* 2010,122:361–369; *Arterioscler Thromb Vasc Biol.* 2011,31:1124-1132]. However, CRTH2 deletion had no markedly influences on expression of CHOP and its target gene GADD34 in infarcted hearts and cultured cardiomyocytes in response to anoxia. Please see Reviewer Figure 1 (not included in this Peer Review file).

2. Additional evidence, such as apoptotic DNA ladder or annexin V, would be needed to support the TUNEL data presented in Figure 2.

As you suggested, we performed additional flow cytometry assay using annexin V/PI staining to detect cardiomyocyte apoptosis under anoxia condition. Consistent with the TUNEL results, cardiomyocytes from CRTH2^{-/-} mice had much less apoptosis (Annexin V⁺) rate than that from WT mice. We amended the results; please see Figure EV1C and D, and page 5, line 18-21.

3. The authors would need to evaluate the Calpain activity and isoform expression pattern at MI border area.

Thanks. As requested, we examined the calpain activity and its isoform expression at MI border area in mice, and found that calpain activity was significantly decreased in CRTH2^{-/-} mice compared to WT mice (Figure EV5A), and m-calpain, u-calpain and calpain 7 were dominant isoforms at MI border area (Figure EV5C). We amended the results; please see page 8, line 19-20, 24-25.

4. The authors would need to discuss the differences between μ-calpain, m-calpain and calpain 7.

We discussed, please see Page 11, Line 24-30; Page 12, Line 1-7.

Referee #3 (Comments on Novelty/Model System):

I found the study well performed and the experimental approach was appropriate to address the hypothesis proposed. The data reported on the involvement CRTH2 activation in cardiomyocyte apoptosis are convincing. However, it would be important to verify whether CRTH2 deletion or blockage may inhibit cardiac fibrosis.

Thank you.

Referee #3 (Remarks):

In the present manuscript, Zuo et al. aimed to study the role of (CRTH2) in cardiac recovery in mice post myocardial infarction and doxorubicin challenge. In particular, they examined the impact of CRTH2 in cardiomyocyte apoptosis. The experimental strategy was to study cardiomyocytes in response to anoxia and DOX treatment in vitro; then they assessed the influence of CRTH2 deletion on Mouse MI model. They found that CRTH2 inhibition protects against MI by reducing ischemia-induced apoptosis in mice. CRTH2 activation promoted anoxia-induced apoptosis in myocytes by m-calpain activation. CRTH2 activation promoted ER stress-induced cardiomyocyte apoptosis through the m-calpain/caspase-12 signaling pathway. In AC16 human cardiomyocytes under anoxia condition, CRTH2 activation promoted caspase-4 activity (caspase-4 the most homologous to mouse caspase-12) along with increased caspase-3 activity. Caspase-4 silence attenuated the increased caspase-3 activity in DK-PGD2-treated AC16 cells. Altogether the findings of this study suggest that the stimulation of CRTH2 receptor exerts a pro-apoptotic effect in cardiomyocytes via the calpain/caspase-12 signaling pathway. They propose that the inhibition of CRTH2 may have therapeutic potential for apoptotic cardiomyopathy.

This is a clinically relevant study using appropriate experimental technologies to address the role of CRTH2 in progressive cardiomyocyte loss associated with ischemic heart disease and chemotherapy-induced cardiomyopathy. Interestingly, the results of this study suggest the possible use of selective antagonists of CRTH2 to protect hearts from the consequences of MI. The manuscript is well and clearly written.

Thank you.

Major points

In response to MI, different reparative responses are induced and importantly the fibrotic responses in the damaged heart. Thus, it would be important to know whether, in addition to an anti-apoptotic effect on cardiomyocytes, CRTH2 deletion or blockage might inhibit cardiac fibrosis. Interestingly, CRTH2 is also expressed in fibroblasts.

Thank you so much, this is a very good question. As suggested, we examined cardiac fibrosis by using both Masson trichrome (Figure 2K, L) and Sirius Red staining (Reviewer figure 2, not included in this Peer Review file). We observed less collagen deposition in the border zones of infarcted hearts in CRTH2^{-/-} mice than that in WT mice at day 14 after MI. As such, we amended the results, please see Page 5, line29-30.

If possible, it would be very interesting to know whether the CRTH2 deletion in cardiac fibroblasts may be associated to reprogramming of fibroblasts into cardiomyocytes.

It is a nice suggestion. Functional cardiomyocytes can be reprogrammed from fibroblasts by three transcriptional factors- Gata4, Mef2c, Tbx5 (GMT), [*Cell.2010;142(3):375-86*]; [*Circ Res. 2015;116(2):237-44*]. By using GMT system, we investigated the impact of CRTH2 deficiency on reprogramming of cardiac fibroblasts into cardiomyocytes by collaborating with Dr. Li Qian (University of North Carolina at Chapel Hill). Immunostaining revealed that cardiac Troponin T (cTnT) positive cells (~15%) were induced from cardiac fibroblasts by GMT, but no significant difference of reprogramming efficiency was detected between WT and CRTH2^{-/-} fibroblasts (Figure EV3A,B); We also observed similar beating rates of cTnT positive cells transdifferentiated from

WT and CRTH2^{-/-} fibroblasts at different timepoints (Figure EV3C). Consistently, similar expression levels of cardiomyocyte-specific genes were induced in WT and CRTH2^{-/-} fibroblasts by GMT transduction (Figure EV3D). These results suggested that CRTH2 is not involved in cardiac reprogramming from fibroblast. Thus, we expanded the methods and results, and cited the relevant references. Please see Page6, line10-20; page 19, line3-22.

2nd Editorial Decision

30 November 2017

Thank you for the submission of your revised manuscript to EMBO Molecular Medicine. We have now received the enclosed reports from the referees that were asked to re-assess it. As you will see the reviewers are now supportive and I am pleased to inform you that we will be able to accept your manuscript pending editorial amendments.

***** Reviewer's comments *****

Referee #1 (Remarks for Author):

None.

Referee #3 (Comments on Novelty/Model System for Author):

The revised version is improved, and this study is of interest to the readers of the Journal. The findings are novel and original and of potential clinical importance.

Referee #3 (Remarks for Author):

The authors replied appropriately to the Reviewers' comments. They performed additional experiments suggested by the Reviewers. The revised version is improved.

Corresponding Author Name: Ying Yu and Yujun Shen

Manuscript Number: EMM-2017-08237